# Pediatric Patients with Stage IV Rhabdomyosarcoma Significantly Benefit from Long-Term Maintenance Therapy: Results of the CWS-IV 2002 and the CWS DOK IV 2004-Trials

**DOI:** 10.3390/cancers15072050

**Published:** 2023-03-30

**Authors:** Lars Tramsen, Konrad Bochennek, Monika Sparber-Sauer, Emilia Salzmann-Manrique, Monika Scheer, Tobias Dantonello, Arndt Borkhardt, Uta Dirksen, Anne Thorwarth, Jeanette Greiner, Martin Ebinger, Jadwiga Weclawek-Tompol, Ruth Ladenstein, Gustaf Ljungman, Erika Hallmen, Thomas Lehrnbecher, Ewa Koscielniak, Thomas Klingebiel

**Affiliations:** 1Department of Pediatrics, University Hospital Schleswig-Holstein, Campus Kiel, 24105 Kiel, Germany; 2Department for Children and Adolescents, University Hospital, Goethe-University, 60590 Frankfurt, Germany; 3Center for Pediatric, Adolescent and Women’s Medicine, Pediatric 5 (Oncology, Hematology, Immunology), Hospital of the State Capital Stuttgart, Olgahospital, Stuttgart Cancer Center, 70174 Stuttgart, Germany; 4Faculty of Medicine, University Tuebingen, 72016 Tuebingen, Germany; 5Department of Pediatric Oncology and Hematology, Charité–Universitätsmedizin Berlin, Augustenburger Platz 1, 13353 Berlin, Germany; 6Department of Pediatrics, Division of Pediatric Hematology and Oncology, Inselspital, Bern University Hospital, University of Bern, CH-3010 Bern, Switzerland; 7Department of Pediatric Oncology, Hematology and Clinical Immunology, Medical Faculty, Heinrich Heine University, 40225 Duesseldorf, Germany; 8Pediatrics III, University Hospital Essen, West German Cancer Center, 45147 Essen, Germany; 9German Cancer Consortium site Essen, National Center for Tumor Diseases (NCT) Site Essen, 45147 Essen, Germany; 10Pediatric Oncology and Hematology, Children’s Hospital, Kantonsspital Aarau AG, CH-5001 Aarau, Switzerland; 11Department of General Pediatrics and Pediatric Oncology and Hematology, University Children’s Hospital, 72076 Tuebingen, Germany; 12Department of Bone Marrow Transplantation, Pediatric Oncology and Haematology, University of Medicine Wroclaw, 50556 Wroclaw, Poland; 13St. Anna-Kinderspital, Children’s Cancer Research Institute (CCRI), 1090 Vienna, Austria; 14Department of Women’s and Children’s Health, Pediatric Oncology, Uppsala University, 75185 Uppsala, Sweden

**Keywords:** soft tissue sarcoma, rhabdomyosarcoma, metastatic, children, long-term maintenance therapy, outcome

## Abstract

**Simple Summary:**

Patients with a metastatic rhabdomyosarcoma treated with a long-term maintenance therapy (LTMT) as a final additional treatment element show a similar outcome compared to a final high-dose chemotherapy regimen and a significantly better outcome compared to patients receiving an allogeneic hematopoietic stem cell transplantation. Particularly, special subgroups such as the very high-risk group defined by alveolar RMS and an Oberlin risk score of ≥ 2 seem to benefit from the LTMT, as they exhibit an overall survival of 37%, which has not been shown before.

**Abstract:**

Rhabdomyosarcoma (RMS) is the most common soft tissue sarcoma (STS) in childhood. Whereas more than 90% of patients with localized low-risk RMS can be cured, metastatic RMS have a dismal outcome, with survival rates of less than 30%. The HD CWS-96 trial showed an improved outcome for patients receiving maintenance therapy after completing intensive chemotherapy. Consequently, the international clinical trials CWS-IV 2002 and CWS DOK IV 2004 on metastatic disease of STS of the Cooperative Weichteilsarkom Studiengruppe (CWS) were designed in addition to the CWS-2002P trial for localized RMS disease. All patients received a multimodal intensive treatment regimen. To maintain remission, three options were compared: long-term maintenance therapy (LTMT) versus allogeneic hematopoietic stem cell transplantation (alloHSCT) versus high-dose chemotherapy (HDCT). A total of 176 pediatric patients with a histologically confirmed diagnosis of metastatic RMS or RMS-like tumor were included. A total of 89 patients receiving LTML showed a significantly better outcome, with an event-free survival (EFS) of 41% and an overall survival (OS) of 53%, than alloHSCT (*n* = 21, EFS 19%, *p* = 0.02, OS 24%, *p* = 0.002). The outcome of LTML was slightly improved compared to HDCT (*n* = 13, EFS 35%, OS 34%). In conclusion, our data suggest that in patients suffering from metastatic RMS, long-term maintenance therapy is a superior strategy in terms of EFS and OS compared to alloHSCT. EFS and OS of HDCT are similar in these strategies; however, the therapeutic burden of LTMT is much lower.

## 1. Introduction

Rhabdomyosarcoma (RMS) is the most common soft tissue sarcoma (STS) in children. STS has an incidence of 0.9 per 100,000 in children under the age of 15 years and consists in 70% of RMS [1,2]. Whereas more than 90% of patients with a localized low-risk RMS can be cured [3,4,5], metastatic, very high-risk RMS have a poor prognosis, with a 5-year overall survival (OS) rate of less than 30% [6,7,8]. For example, patients suffering from alveolar RMS or RMS with two or more nowadays called Oberlin risk factors, including age (≤1 year or ≥10 years), multiple metastatic sites (≥3), bone or bone marrow involvement, or unfavorable primary site, have a 3-year event-free survival (EFS) of 14% only [9].

While there have been only a few attempts with allogeneic hematopoietic stem cell transplantation (alloHSCT) to date, several studies tried to improve the outcome in metastatic RMS using high-dose chemotherapy (HDCT) [10,11,12,13]. In the ARST0431 study of the Children’s Oncology Group, interval compression was used for therapy intensification [14].

Unfortunately, the use of treatment modalities such as alloHSCT or HDCT followed by autologous stem cell rescue failed to improve overall outcome in patients suffering from metastatic STS [14,15,16,17,18].

The prospective clinical study “HD CWS-96 on metastatic disease” compared high-dose therapy with oral maintenance treatment in a nonrandomized fashion in 96 pediatric patients with metastatic STS. A distribution analysis showed no differences in prognostic factors between the two groups. The HD CWS-96 trial resulted in an OS of 52% in the oral maintenance treatment group compared to 27% in the high-dose therapy group [15]. The subsequent studies CWS-IV 2002 and CWS DOK IV 2004 on metastatic disease included long-term maintenance therapy (LTMT) up to 24 months at the end of intensive treatment for patients with metastatic disease with trofosfamide and etoposide alternating with trofosfamide and idarubicin (O-TIE) [15] or maintenance therapy using oral cyclophosphamide and intravenous vinblastine (CYC/VBL). CYC/VBL was first introduced in the CWS-2002P study for high-risk patients [4]. A similar maintenance chemotherapy with vinorelbine and low-dose cyclophosphamide for treatment of patients with nonmetastatic high-risk RMS was included after a pilot study [19] into the RMS 2005 trial of the European pediatric Soft tissue sarcoma Study Group (EpSSG) and seemed to improve the outcome of these patients [20,21]. The MTS 2008 Study for patients with metastatic RMS performed by the EpSSG also included a maintenance therapy with vinorelbine and low-dose cyclophosphamide. In pooled analysis with the concurrent BERNI Study, the outcome of patients with ≥ 3 Oberlin risk factors had an improved 3-year OS of 26% [22,23]. However, whether a metronomic LTMT after conventional intensive polychemotherapy and local treatment might be effective or a low intensive LTMT might be the more adequate choice for patients suffering metastatic RMS remained unclear.

We report prognostic factors and outcome of patients receiving multimodal treatment for RMS followed by either LTMT, HDCT, or alloHSCT. In addition, we compared the prognosis of patients who received either O-TIE or CYC/VBL as LTMT. The treatment decisions are based on the choice of the treating attending physicians.

## 2. Patients and Methods

### 2.1. Eligible Patients

A total of 176 patients with metastatic RMS and RMS-like STS (stage IV) diagnosed between November 2002 and July 2010 were enrolled in the CWS-IV 2002 and the CWS DOK IV 2004 trials of the German Society of Pediatric Oncology and Hematology (GPOH). Trial inclusion criteria were histologically confirmed diagnosis of the STS entities RMS (*n* = 151) and RMS-like tumors. Patients with RMS-like tumors had the diagnosis extraosseous Ewing’s sarcoma (*n* = 11), synovial sarcoma (*n* = 9), and undifferentiated sarcoma (*n* = 5). Further inclusion criteria were age younger than 21 years, no pretreatment (chemotherapy or radiation therapy), and no previous malignant disease. Patients were treated in 64 Pediatric Cancer Centers in Germany, Sweden, Switzerland, Poland, and Austria between November 2002 and July 2010. The median follow-up of survivors was 5.8 years (11 months to 9 years). The study protocols were approved by the ethical committee of each participating center, and written informed consent was obtained by patient and/or caregiver.

### 2.2. Treatment Schedule According to CWS-IV 2002 and CWS 2002 Doku-Trial

All patients received polychemotherapy according to the protocol CWS-IV 2002 (*n* = 31) or CWS DOK IV 2004 (*n* =145), including surgery or radiotherapy, respectively. The protocol CWS-IV 2002 was a prospective nonrandomized phase II multicenter cohort study, starting with a window of two courses of topotecan and carboplatin (TC). The aim of the window trial was to evaluate response and toxicity of this combination compared to monotherapeutic topotecan therapies published before. The window trial did not result in improved outcome [24]. Patients with good response after the window received two consecutive cycles of ifosfamide, vincristine, actinomycin D (I^3^VA), ifosfamide, vincristine, adriamycin (I^3^VAd), and TC. Intensive chemotherapy was completed by one additional I^3^VA course. All patients who did not respond to the window regimen received seven courses of chemotherapy, including I^3^VA, alternating with carboplatin, epirubicin, vincristine (CEV) and ifosfamide, vincristine, and etoposide (I^3^VE). In case of poor response after three courses of chemotherapy, treatment was continued off-study for up to 49 weeks after initial diagnosis.

A total of 14 patients received O-TIE after week 25 for up to 24 weeks. O-TIE therapy consisted of trofosfamide (2 × 75 mg/m^2^/d) and etoposide (2 × 25 mg/m^2^/d) alternating with trofosfamide (2 × 75 mg/m^2^/d) and idarubicine (1 × 5 mg on day one). Each course was administered continuously for 10 days, with a break of 10 days in between. Local therapy was performed between week 7 and 10 according to the protocol CWS-96 for localized disease.

The majority of patients in protocol CWS DOK IV 2004 received nine alternating courses of I^3^VA, CEV, and I^3^VE (CEVAIE; *n* = 68). A second group in this protocol (31 patients) was treated with VAIA III, which included five courses of I^2^VAd^2^ and four courses of I^2^VA. 

Response assessment was performed at week nine using magnetic resonance imaging, and patients with a poor response defined as reduction in tumor volume of less than 50% switched to a second-line treatment consisting of six alternating courses of TC, topotecan, and cyclophosphamide and carboplatin and etoposide. A third group consisting of 36 patients received a therapy according to the phase II window trial described above (*n* = 36). Ten patients were treated with an individual chemotherapeutic regimen at an early stage.

The majority of patients of the CWS DOK IV 2004 study received LTMT for up to 24 weeks with O-TIE (*n* = 57) or an alternative LTMT with cyclophosphamide (2 × 25 mg/m^2^/d given orally) and vinblastine (3 mg/m^2^ given once a week intravenously) (CYC/VBL; *n* = 17) administered continuously for three weeks, followed by a one-week break. Only one patient of the CWS-IV 2002 trial also received CYC/VBL.

A total of 21 patients underwent alloHSCT after intensive chemotherapy, and 13 patients received HDCT consisting of one course of thiotepa/cyclophosphamide and one course of melphalan/etoposide with autologous stem cell rescue, which was at the discretion of the treating physician. These patients did not receive oral maintenance therapy (Figure 1).

In both studies, treatment allocation was not determined by randomization but was based on the decision of the treating attending physicians. 

Toxicity data were defined according to the CTCTAE grading system. Data were collected for the whole course of therapy in each participating patient; therefore, no sub analysis of single therapeutic elements was possible, since adverse events (AEs) were reported and summarized for the whole therapy course but not assigned to the single-therapy elements.

### 2.3. Statistical Methods

Descriptive statistics are presented as absolute value and percentages or median with range for categorized and quantitative variables. Patients’ characteristics variables were compared using Fisher’s exact test for categorical variables and Wilcoxon–Mann–Whitney tests for continuous variables. Median follow-up time of survival was obtained using the reverse Kaplan–Meier method. To estimate the probabilities of OS and EFS for all patients, the observation time was calculated from start of the initial chemotherapy to the respective event or last follow-up for censored patients. OS was defined as number of survivors minus the number of patients who died, independently of the cause of death. Events EFS were defined as disease progression or relapse, second malignancy, and non-relapse mortality, respectively. Non-relapse mortality was defined as death without prior relapse or progression. To estimate the probabilities of OS and EFS in patients receiving LTMT, HDCT, or alloHSCT, the observation time was calculated from the start of one of these three final treatment approaches. Probabilities of OS and EFS were estimated by the Kaplan–Meier product limit method. Univariate and multivariate comparisons of OS and EFS probabilities were performed using the Log-rank test. The association between patients’ characteristics and outcomes was analyzed using Cox proportional hazard models.

All statistical tests were 2-sided, and *p* values < 0.05 were considered significant. Analyses were performed using the statistical software R, R-Core-Team (The R Foundation for Statistical Computing, University of Vienna, Austria, 2019).

## 3. Results

### 3.1. Patient Characteristics

A total of 123 patients with metastatic RMS diagnosed between November 2002 and July 2010 were included in the analysis. In total, 53 patients were excluded from detailed analysis because of early discontinuation of treatment (*n* = 15), receiving individual chemotherapy (*n* = 20), or they did not receive a final LTMT, HDCT with autologous stem cell rescue, or alloHSCT (*n* = 18), respectively (Figure 1).

The patients´ characteristics are summarized in Table 1. 

The median (range) age at time of diagnosis was 11.9 years (1 month to 21 years). According to prognostic risk factors for metastatic disease defined previously [5], 69 patients (39%) were in the favorable age group (1 to 9 years) and 107 patients (61%) were in the unfavorable age group (10 years and older (*n* = 103) or younger than 1 year (*n* = 4), respectively). A total of 161 patients (91%) had an unfavorable primary site of disease as described in Table 1, 55 patients (31%) had three or more metastatic sites, and 84 patients (48%) had bone marrow involvement or bone metastases, respectively.

The median (range) follow-up time was 121 weeks (1 to 278 weeks). Estimated 3-year EFS and OS for all patients were 31% (95% CI, 25 to 39%) and 45% (95% CI, 38 to 63%), respectively (Figure 2).

With respect to the three patient groups analyzed in detail (LTMT, HDCT, and an alloHSCT), a proportion of 67%, 92%, and 95%, respectively, had an Oberlin prognostic score of ≥ 2 (Appendix A).

### 3.2. Outcome of Patients Receiving LTMT

Out of all patients, 89 (51%) received LTMT, with a 3-year EFS and OS of 41% (95% CI, 31 to 52%) and 53% (95% CI, 43 to 64%), respectively (Figure 3).

No significant differences of EFS or OS were seen in patients of the LTMT group with respect to the different initial chemotherapy approaches using Window TC/VAIA versus CEVAIE (Appendix A). Comparing the two different LTMT regimes O-TIE (*n* = 71) and CYC/VBL (*n* = 18), patients receiving CYC/VBL showed a tendency towards better EFS and OS of 60% (95% CI, 40 to 88%) and 70% (95% CI, 52 to 96%), respectively, compared to those receiving O-TIE (EFS, 36% (95% CI, 26 to 49%); OS, 48% (95% CI, 38 to 62%)), although these differences did not reach statistical significance (OS *p* = 0.08, EFS *p* = 0.09) (Figure 4).

When dividing analyzed LTMT patients into two subgroups according to the Oberlin prognostic score, patients with a favorable prognostic score of ≤1 showed a 3-year EFS and OS of 55% (95% CI, 40 to 77%) and 65% (95% CI, 50 to 85%), respectively. In contrast, patients with a score of ≥2 had a poor outcome with a 3-year EFS and OS of 34% (95% CI, 23 to 48%) and 46% (95% CI, 35 to 61%), respectively (Figure 5).

Considering the histological subtypes, patients in the LTMT group with an embryonal RMS had the best outcome, with a 3-year EFS and OS of 56% (95% CI, 41 to 77%) and 73% (95% CI, 59 to 91%), respectively. Patients suffering from alveolar RMS exhibited an EFS and OS of 29% (95% CI, 18 to 46%) and 38% (95% CI, 26 to 56%) (Appendix A). Patients with an alveolar RMS and an Oberlin risk score of ≥ 2 had the worst outcome, with a 3-year EFS and OS of 17% (95% CI, 11 to 28%) and 28% (95% CI, 19 to 40%), respectively (Figure 6A,B). However, EFS and OS of patients in this subgroup were slightly better with 29% (95% CI, 17 to 49%) and 37% (95% CI, 24 to 57%) when they received LTMT (Figure 6C,D).

### 3.3. Outcome Depending on Treatment Strategies after Intensive Chemotherapy

Patients receiving an LTMT had significantly (*p* = 0.02 and 0.002) better EFS and OS (41% (95% CI, 32 to 52%) and 53% (95% CI, 43 to 64%), respectively)) compared to patients finally treated with alloHSCT (EFS of 19%; 95% CI, 8 to 46%; OS of 24%; 95% CI, 11 to 51%). Results of patients receiving HDCT were similar to LTMT (EFS of 35%; 95% CI, 16 to 76%; OS of 34%; 95% CI, 16 to 75%) (Figure 7).

### 3.4. Toxicity 

All patients included in this study had received intensive poly-chemotherapy as first line therapy, in many cases combined with radiotherapy and/or surgery. Adverse events of CTCTAE grade 3 and 4 toxicity were common, with hematotoxic AEs > grade 2 occurring in 145 patients (82%) and skin and mucosal AEs > grade 2 in 148 (84%) patients. Adverse events of the liver > grade 2 was seen in 35% of the patents, infections, cardiotoxicity, and gastrointestinal AEs < grade 2 occurred in 25%. There was less toxicity observed in these categories for patients receiving LTMT compared to patients treated with alloHSCT or HDCT. However, the differences were not statistically significant. In contrast, CNS toxicity and pain grade >2 were reported significantly more often in patients receiving alloHSCT than in those receiving maintenance therapy (*p* = 0.014 and *p* = 0.049, respectively).

### 3.5. Multivariate Analysis

In the multivariate analysis, age above 10 years (*p* = 0.016), unfavorable tumor site (*p* = 0.031), bone marrow involvement (*p* = 0.036) and three or more metastatic sited (*p* = 0.039) could be identified as independent risk factors for lower OS, relating to the whole course of therapy. For EFS, only age (*p* = 0.009) and site (*p* = 0.046) were independent risk factors. When analyzing LTMT only, age (*p* = 0.03) and alveolar subtype (*p* = 0.015, *p* = 0.009 respectively) were independent risk factors for a worse outcome regarding OS and EFS, while bone marrow involvement was a risk factor for OS only (*p* = 0.009).

## 4. Discussion

Metastatic RMS are still a therapeutic challenge and, unfortunately, most novel therapeutic approaches did not improve the poor outcome in these patients. Data in a previous nonrandomized trial suggested a significantly better outcome for high-risk patients receiving oral maintenance treatment when compared to high-dose therapy [15]. Our study of 176 patients suffering from RMS and RMS-like metastatic STS could show a significantly better outcome for patients treated with LTMT as a final additional treatment element (*p* = 0.008) when compared to patients receiving alloHSCT. Results of LTMT were similar to HDCT treatment. Special subgroups such as the very high-risk group defined by an alveolar RMS and an Oberlin risk score of ≥ 2 seem to benefit from the LTMT, as they exhibit an OS of 37%, which has not been shown before.

Recent studies showed the importance of molecular markers, especially the presence or absence of FOXO1 fusion status for classification of RMS risk groups [25,26,27]. Such additional information might be suitable for assessment of different final treatment elements, such as LTMT, HDCT, or alloHSCT [28]. For our present analysis these molecular data were not completely available.

Most high-dose therapy regimens require an autologous stem cell rescue due to bone marrow toxicity, which, in turn, leads to more treatment-associated adverse events, such as infectious complications. Unfortunately, intensive treatment regimens did not result in higher overall survival [29]. In 2016, the Children’s Oncology Group presented a different high-dose therapy approach using dose intensification by interval compression [14]. With this strategy, patients having Oberlin score ≤ 1 had a 3-year EFS of 67% (95% CI, 50% to 79%), whereas very high-risk patients (Oberlin score ≥2) still had a very poor outcome (EFS 19%; 95% CI, 10% to 30%), which is in line with the results of the present analysis [9].

LTMT offers a similar outcome compared to high-dose treatment reported by Weigel et al. 2016 [14] and, importantly, may also be associated with higher quality of life, since LTMT allows an outpatient setting and is associated with a lower hospitalization rate, with no need for invasive medical procedures such as the use of a central venous line and with less toxicity.

Unfortunately, in our study, toxicity data were summarized and, therefore, available for the whole course of therapy only, but, overall, patients with stage IV STS receiving intensive chemotherapy followed by alloHSCT suffered more from heavy pain and CNS adverse events compared to patients receiving intensive chemotherapy followed by LTMT.

Our data also demonstrate that alloHSCT did not improve outcome, which corroborates previous studies [16,17,30]. However, it is important to note that data on alloHSCT are based on nonrandomized studies only, including very small patient numbers. Our preliminary data in a small number of patients suggest that LTMT using CYC/VBL (Oberlin score comparable in both groups) results in a better outcome than O-TIE, although the data did not reach statistical significance (OS 70% vs. 48%, *p* = 0.08; EFS 60% vs. 36%, *p* = 0.09). Due to the limited number of patients, we are not able to draw final conclusions on the best maintenance strategy. In this context, it is important to note that the results achieved by LTMT were independent from the different conventional chemotherapy approaches used during intensive polychemotherapy (Window TC/VAIA vs CEVAIE).

Recently, it was reported that metronomic maintenance treatment using cyclophosphamide/vinorelbine in a randomized trial, which is similar to CYC/VBL given in our study, improved survival in 371 patients suffering from high-risk rhabdomyosarcoma (OS 86.5% with LTMT vs. 73.7 without *p* = 0.009) [20]. However, these data are on localized RMS disease. The BERNIE study from 2017 evaluated the randomized, additional use of bevacizumab in childhood and adolescent patients with metastatic STS, also including a maintenance therapy using cyclophosphamide/vinorelbine. The treatment regime without bevacizumab was comparable to our study. EFS in patients with metastatic RMS was 36.0% (95% CI: 25.2–47.9) [22] and, thereby, similar to our data.

We recognize that our analysis has significant limitations. For example, treatment allocation was not determined by randomization but was based on the decision of the treating attending physicians. As a result, more patients with higher risk score received a more intensive therapy (Oberlin score ≥ 2: 67% in LTMT vs. 95% in alloHSCT, Appendix A), which clearly is a treatment bias. We strongly support future European randomized trials to gain reliable data on RMS and RMS-like STS, which are very rare tumor entities in childhood.

Another limitation of this analysis is the lacking data on toxicity of different parts of systemic treatments. Consequently, we are not able to report data on toxicity of LTMT versus HDCT or alloHSCT separate from the initial multimodal intensive treatment regimen. However, pain and neurological problems were much more common in transplantation treatment than in the maintenance setting. 

## 5. Conclusions

In conclusion, our data report that LTMT is a superior strategy to alloHSCT in patients suffering from metastatic RMS. This LTMT regimen significantly improved the outcome of the very high-risk group defined by an alveolar RMS and an Oberlin risk score of ≥ 2, as they exhibit an OS of 37%, which has not been shown before for any kind of therapy. Compared to HDCT, EFS und OS was similar but with less therapeutic burden. LTMT had an acceptable toxicity profile and did not report pain and neurological problems, as did HDCT. 

Further controlled clinical trials are needed to evaluate different regimens of LTMT, such as O-TIE versus CYC/VBL, the duration of LTMT, and specific toxicity.

## Figures and Tables

**Figure 1 cancers-15-02050-f001:**
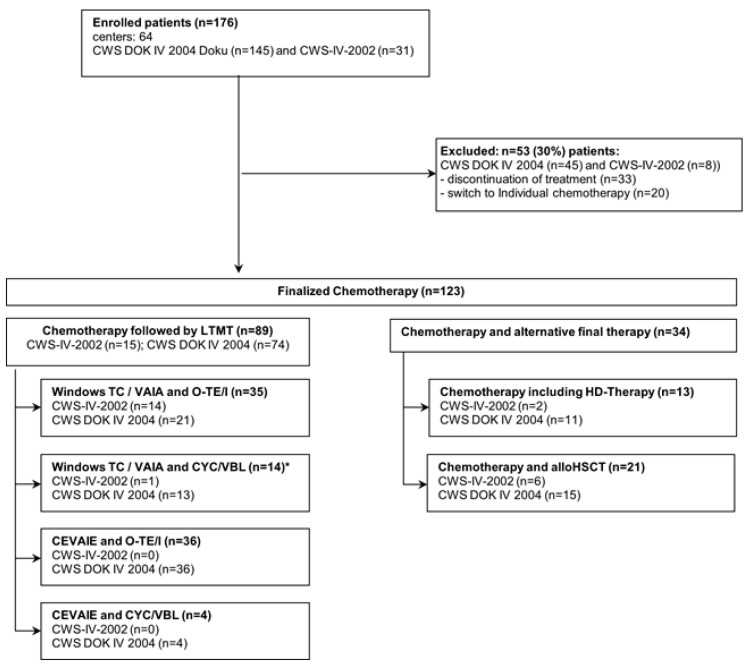
The diagram illustrates the distribution of 176 patients on inclusion and exclusion criteria, and the different chemotherapy treatment regimens of all patients. * One patient with other therapy but including also WTC/VAIA + CYC/VBL was allocated here.

**Figure 2 cancers-15-02050-f002:**
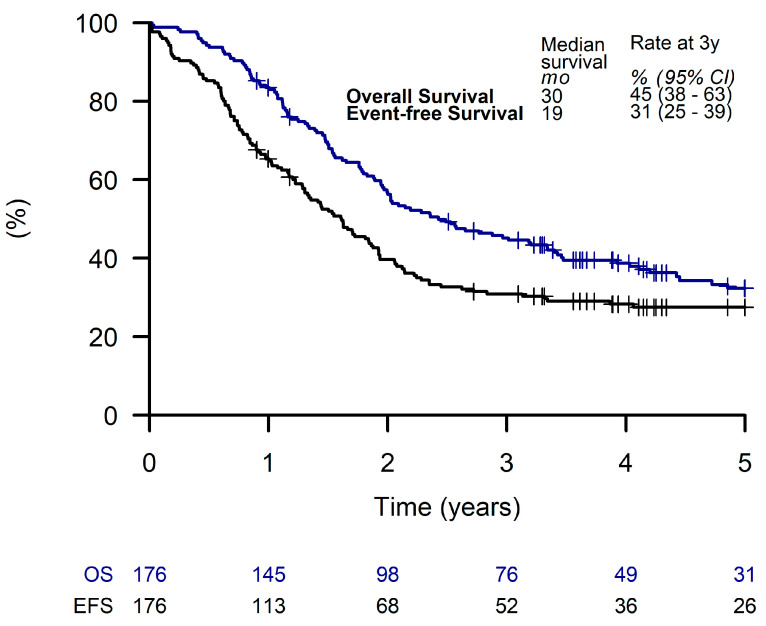
Overall survival and event-free survival of all patients.

**Figure 3 cancers-15-02050-f003:**
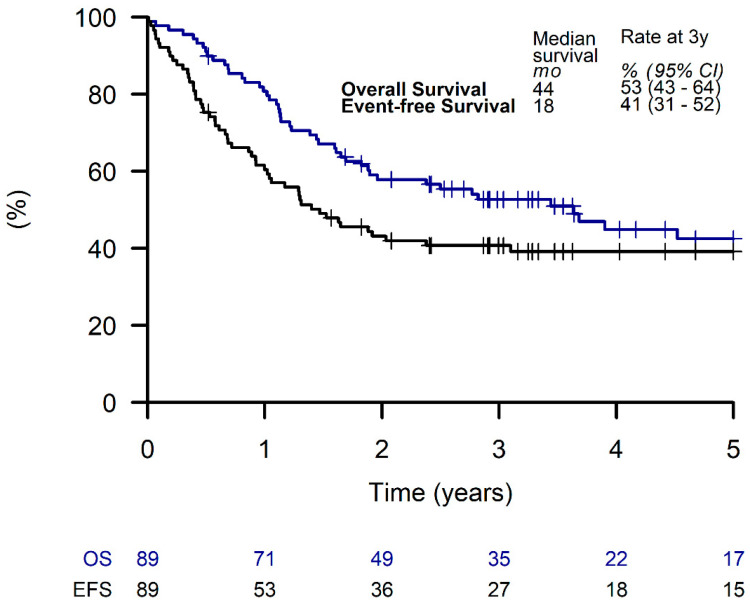
Overall survival and event-free survival of patients receiving a metronomic long-term maintenance therapy.

**Figure 4 cancers-15-02050-f004:**
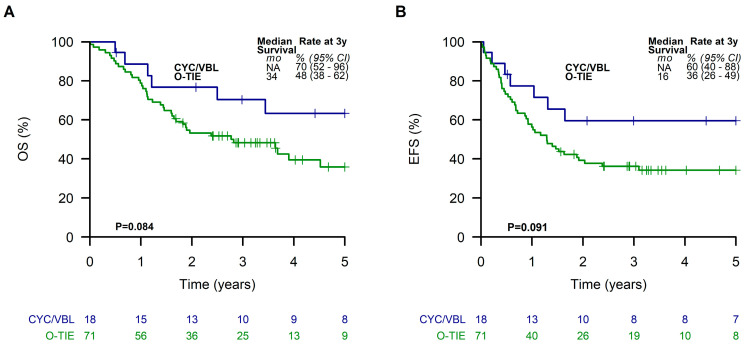
Overall survival (**A**) and event-free survival (**B**) of patients receiving the two different metronomic long-term maintenance therapies.

**Figure 5 cancers-15-02050-f005:**
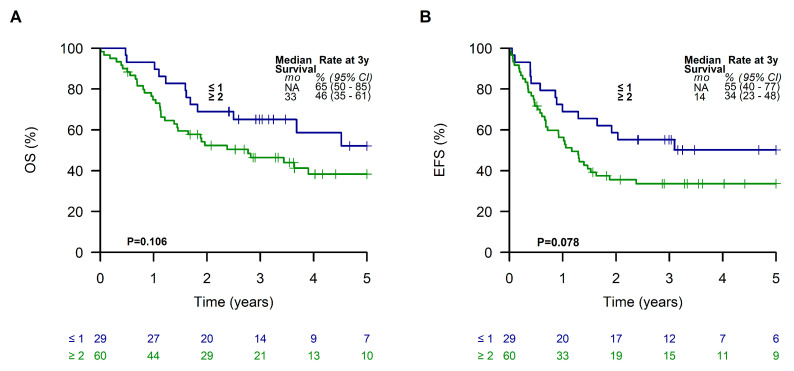
Overall survival (**A**) and event-free survival (**B**) of patients receiving a metronomic long-term maintenance therapy according to the Oberlin risk groups.

**Figure 6 cancers-15-02050-f006:**
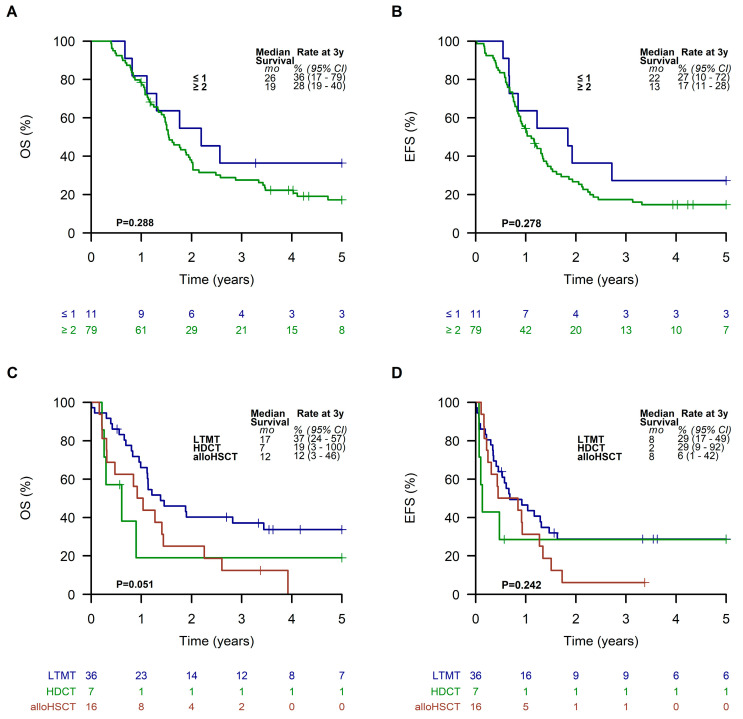
Overall survival (**A**) and event-free survival (**B**) of all patients with an alveolar RMS and an Oberlin risk score of ≥2. Overall survival (**C**) and event-free survival (**D**) of patients with an alveolar RMS and an Oberlin risk score of ≥2 receiving a metronomic long-term maintenance therapy (LTMT), high-dose chemotherapy (HDCT), or allogeneic hematopoietic stem cell transplantation (alloHSCT).

**Figure 7 cancers-15-02050-f007:**
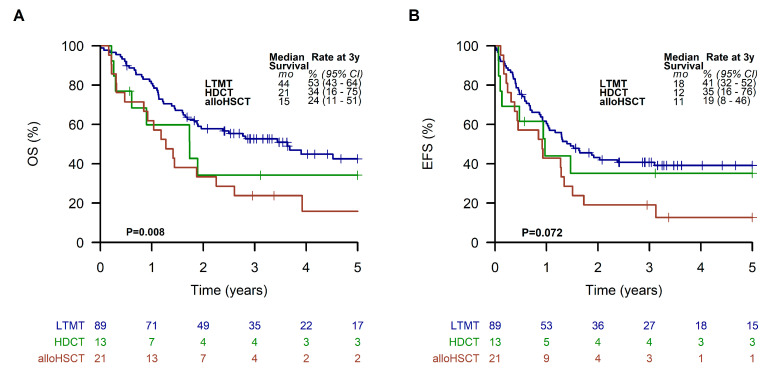
Overall survival (**A**) and event-free survival (**B**) of all patients receiving a metronomic long-term maintenance therapy (LTMT), allogeneic hematopoietic stem cell transplantation (alloHSCT), or high-dose chemotherapy (HDCT).

**Table 1 cancers-15-02050-t001:** Patient characteristics by initial treatment protocol.

Patient Characteristics				
	CWS-IV 2002	CWS DOK IV 2004	Total	*p*
N (%)	31 (18)	145 (82)	176 (100)	
**Sex, N (%)**				0.825
Female	15 (48)	67 (46)	82 (47)	
Male	16 (52)	78 (54)	94 (53)	
**Age, years**				
Median (range)	15 (2–21)	12 (0–20)	12 (0 -21)	0.048
**Age, N (%)**				
≤1 y	0	4 (3)	4 (2)	0.792
1–9	11 (35)	58 (40)	69 (39)	
≥10	20 (65)	83 (57)	103 (59)	
**Histology, N (%)**				0.617
Embryonal RMS	8 (26)	50 (35)	58 (33)	
Alveolar RMS	17 (55)	73 (50)	90 (51)	
Other	6 (19)	22 (15)	28 (16)	
**Primary site of tumor, N (%) ***				0.800
Favorable	3 (10)	12 (8)	15 (9)	
Unfavorable	28 (90)	133 (92)	161 (91)	
**Tumor size, N (%)**				0.730
≤5 cm	5 (16)	25 (17)	30 (17)	
>5 cm	25 (81)	110 (76)	135 (77)	
Unknown	1 (3)	10 (7)	11 (6)	
**Number of metastases, N (%)**				1
≤2	21 (68)	100 (69)	121 (69)	
≥3	10 (32)	45 (31)	55 (31)	
**Bone or bone marrow involvement, N (%)**				0.431
Yes	17 (55)	67 (46)	84 (48)	
No	14 (45)	78 (54)	92 (52)	
**Oberlin score, (%)**				0.822
≤1	7 (23)	38 (26)	45 (26)	
≥2	24 (77)	107 (74)	131 (74)	
**Period of start initial chemotherapy**				0.294
Median	November 2006	December 2005	June 2006	
Range	February 2005–December 2007	November 2002–July 2010	November 2002–July 2010	
**Initial chemotherapy, N (%)**				<0.001
Window TC/VAIA	31 (100)	70 (48)	101 (57)	
CEVAIE	0	69 (48)	69 (39)	
Others	0	6 (4)	6 (3)	
**Time between start initial chemotherapy and final treatment, days**				0.431
Median	252	254	254	
Range	133–348	167–472	133–472	
Excluded	8	45	53	
**Period of start final treatment**				0.768
Median	July 2007	December 2006	April 2007	
Range	September 2005–Jun 2008	June 2003–May 2011	June 2003–May 2011	
Excluded	8	45	53	
**Final treatment, N (%)**				0.558
LTMT	15 (48)	74 (51)	89 (51)	
HDC	2 (6)	11 (8)	13 (7)	
alloSCT	6 (19)	15 (10)	21 (12)	
Excluded	8 (26)	45 (31)	53 (30)	

* Favorable: orbit, genitourinary non-bladder/prostate (i.e., paratesticular or vagina/uterus), non-parameningeal head and neck. Unfavorable: all other sites (parameningeal, extremities, genito-urinary bladder/prostate, and “other site”). *p* values determined using Fisher’s exact test, chi-square test or Mann–Whitney U test as appropriate comparing the CWS-IV 2002 and the CWS DOK IV 2004 groups.

## Data Availability

Individual participant data are not publicly available since this requirement was not anticipated in the study protocols.

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
