# Peer review of "Pediatric Patients with Stage IV Rhabdomyosarcoma Significantly Benefit from Long-Term Maintenance Therapy: Results of the CWS-IV 2002 and the CWS DOK IV 2004-Trials"

_cancers, 2023, doi:10.3390/cancers15072050_

Round 1
Reviewer 1 Report
The manuscript of Tramsen L. al. demonstrates the benefit of long-term maintenance therapy in stage IV pediatric rhabdomyosarcoma patients. This clinical work is interesting and have direct translational significance. However, the manuscript has some issues that need to be addressed.
MAJOR POINTS
1- The Introduction should be expanded, since it gives very little information, especially about the protocols from other countries and consortia and the treatments that are currently used in general for RMS. For example, the current treatment protocols of the EPSSG and COG should be detailed, which are not even mentioned. Quality clinical research like the one proposed in this work cannot ignore the rest of the protocols that are applied in the world. This is also related to the low number of references in the manuscript, with only 14. There is a lot of background missed and the study should be focused in the general context of the disease.
2- Please review the text throughout the manuscript. Particularly in the Introduction and Abstract, there is an unacceptable number of typos and misspellings. For example, only in the first paragraph of introduction: Line 72: “RhabdomyosarKoma”, “saCRcoma”, two points at the end of the sentence, after “children..”; Line 75: “5 year survival” should be “5-year survival”; Line 76 After the reference [3,4] there is an extra space; And Line 78: “3-years EFS” should be “3-year” in singular. Please, review all the manuscript, particularly Introduction and Abstract.
MINOR POINTS
1- For me it is not clear what is exactly and how is calculated the P-value on the last column on Table 1 (And supplementary Table 1). It should be defined on the Table legend.
2- In figures 6 and 7, is it necessary to include text labels and data within plots overlapping with the lines? I suggest to change this design.
Author Response
Please see the attachment in the box

Reviewer 2 Report
The manuscript entitled, “Pediatric Patients with Stage IV Rhabdomyosarcoma Significantly Benefit from Long Term Maintenance Therapy: Results of the CWS-IV-2002 and the CWS DOK IV 2004-Trials,” is of interest and highlights important information for RMS patients. Here are some suggestions to improve the manuscript:
· Rhabdomyosarcoma is misspelled in abstract line 49 and in the intro line 72
· Introduction could include a bit more detail on the types of RMS, metastatic sites, etc.
· There are lots of abbreviations in the paper which makes it difficult for the readers to remember each one. If possible please minimize the amount of some abbreviations.
· Table 1: Does race or ethnicity of the patient have an impact?
· In Table 1: for alveolar RMS how many patients have fusion positive vs fusion negative cases? Does the level of fusion protein have an impact?
· Figure 6 has “No. of patients, No. of events, Median Survival, Rate at 3Y” all written over the graphs so its hard to read and see the graph.
· The figure legends for supplementary figures should be written under the figures.
· Supplementary Materials where the authors have to list all supplementary tables/ figs only include Table S1. The authors need to revise to include Figure S1 and Figure S2.
· The authors should include more references from previous types of clinical trials that been conducted in RMS. For this type of manuscript 14 references seems insufficient in terms of highlighting what is known in the scientific community.
Author Response
Please see the attachment in the box

Round 2
Reviewer 1 Report
The autors have ammended all my previous concerns. Congratulations for the nice work.